# Direct transfer of tri- and di-fluoroethanol units enabled by radical activation of organosilicon reagents

Xiang Chen[1,2], Xingxing Gong[1,2], Zhengyu Li[1], Gang Zhou[1], Zhihong Zhu[1], Weilu Zhang[1], Shanshan Liu 🄳 [1] & Xiao Shen 🄳 [1✉]

Trifluoroethanol and difluoroethanol units are important motifs in bioactive molecules, but the methods to direct incorporate these units are limited. Herein, we report two organosilicon reagents for the transfer of trifluoroethanol and difluoroethanol units into molecules. Through intramolecular C-Si bond activation by alkoxyl radicals, these reagents were applied in allylation, alkylation and alkenylation reactions, enabling efficient synthesis of various tri(di)fluoromethyl group substituted alcohols. The broad applicability and general utility of the approach are highlighted by late-stage introduction of these fluoroalkyl groups to complex molecules, and the synthesis of antitumor agent Z and its difluoromethyl analog Z'.

[1] The Institute for Advanced Studies, Engineering Research Center of Organosilicon Compounds & Materials, Ministry of Education, Wuhan University, 430072 Wuhan, China. [2] These authors contributed equally: Xiang Chen, Xingxing Gong. ✉email: xiaoshen@whu.edu.cn

Fluorine incorporation has been a routine strategy for the design of new drugs and materials, because it can often improve the chemical, physical, and/or biological properties of organic molecules[1,2]. Fluoroalkylsilicon reagents, such as TMSCF$_3$ (Rupert–Prakash reagent), TMSCF$_2$H, and others are widely used reagents in the synthesis of organofluorine compounds[3]. Among various fluorine-containing molecules, the secondary fluoroalkyl alcohols are of particular importance; monoamine oxidase A inhibitor Befloxatone[4] and antitumor agent **Z**[5] are examples of bioactive molecules containing trifluoroethanol motif (Fig. 1a). Anionic activation of C–Si bond of fluoroalkylsilicon reagents by Lewis bases is a powerful method to transfer α-fluoro carbanions into aldehydes, affording fluoroalkyl alcohol products (Fig. 1b)[6–8]. However, we envisioned that the development of organosilicon reagents such as **1a** and **2a** which allows direct transfer of trifluoroethanol and difluoroethanol into organic molecules would represent a conceptually different means to construct fluoroalkyl alcohols (Fig. 1c). Actually, the synthetic chemistry based on carbonyl group (Fig. 1b) possesses some limitations: (1) many aldehydes are not stable and/or need multistep synthesis[9,10]; (2) it is hard to control the regioselectivity when there are more than one aldehyde sites in the same molecule. Moreover, the design and synthesis of pharmaceuticals call for strategies to incorporate important structural motifs at late-stage, because this will aviod de novo synthesis[11,12].

Herein, we report two fluoroalkylsilicon reagents **1a** and **2a**. Through intramolecular C–Si bond activation by alkoxyl radicals[13–19], these developed β-fluorinated organosilicon reagents were successfully applied in radical allylation, alkylation, and alkenylation reactions, enabling efficient synthesis of a variety of fluoroalkyl group substituted alcohols (Fig. 1c). The broad applicability and general utility of the approach are highlighted by late-stage introduction of fluoroalkyl groups to complex molecules, such as the derivatives from biologically active naturally occuring epiandrosterone, cholesterol, testosterone, diosgenin, vitamin E, estrone, and (8α)-estradiol, and the synthesis of antitumor agent **Z** and its difluoromethyl analog **Z′**. Moreover, our radical reactions show conjunctive group tolerance to that of the traditional nucleophilic fluoroalkylation reactions with α-fluoro carbanions[3,20,21] (Radical reactions often show different reactivity to the anionic reactions, see refs.[20,21]).

## Results

### Preparation of reagents 1a and 2a. The fluoroalkyl group transfer reagents **1a** and **2a** were easily synthesized in three steps (Fig. 2). Following the reported procedure[22], with commercially available inexpensive trifluoroethanol **3** as starting material, we prepared difluorinated enol silyl ether **4** in 84% yield.

Electrophilic fluorination of compound **4** with Selectfluor afforded trifluoroacetylsilane **5** in 82% yield. Reagent **1a** was then synthesized in 86% yield through reduction of acylsilane **5** with NaBH$_4$. We also prepared different silyl group-substituted compounds **1b**–**1d** through similar procedures as **1a** in good yield (for details, see Supplementary Figs. 2–4). It is worthy to note that trifluoroacetyltriphenylsilane (precursor to **1d**) can be synthesized directly from the reaction of Ph$_3$SiLi and (CF$_3$CO)$_2$O in one step[23]. The difluoroacetylsilane **6** was easily prepared in 79% yield through the hydrolysis of enol silyl ether **4** under acidic conditions. Reduction of compound **6** delivered difluoromethyl containing reagent **2a** in 88% yield.

### Attempts for anionic activation and design of radical activation. With reagent **1a** in hand, we investigated the substitution reaction with allylic sulfone **7a** as model substrate. Firstly, we tried anionic activation strategy which has been widely used in the C–Si bond cleavage for the fluoroalkyl transfer reactions[6–8]. Common activators such as TBAF, CsF, KF, *t*-BuOK were tried, but no substitution product **8a** or **9a** was observed, albeit full conversion of compound **1a** was observed (Fig. 3a). The decomposition of compound **1a** could be explained by the facile fluoride elimination of β-fluoro carbanions (Fig. 3b)[23–25]. For example, Xu and coworkers reported the reaction of trifluoroacetyltriphenylsilane with Grignard reagents, but no desired trifluoromethylated alcohols were obtained. Instead, 2,2-difluoro enol silyl ethers were formed through nucleophilic addition, anion Brook rearrangment, and fluoride elimination processes[23]. It is known that fluorine radical possesses much higher energy than fluorine anion does (Fluorine possesses high electron affinity (3.448 eV), extreme ionization energy (17.418 eV), see ref 1a, pp 5–8). Therefore, we envisioned that in-situ generated carbon radical from alkoxyl radicals should not prefer β-F elimination. Consequently, they could be trapped by radical acceptors to generate trifluoroethanol transfer products (Fig. 3b).

### Identification of conditions for radical C–Si activation. With this idea in mind, we investigated a variety of conditions which could generate alkoxyl radicals (for details, see Supplementary Tables 1–3). After extensive screening, we found that employing 2 equivalent of Mn(OAc)$_3$·2H$_2$O as oxidant, DCM as solvent led to allylation product in 68% yield (Table 1, entry 1). Further investigation revealed that with 20 mol% of Mn(OAc)$_3$·2H$_2$O as catalyst and 2.5 equivalent of TBPB as oxidant, product **8** was given in 61% yield (Table 1, entry 2). Changing the silyl group from SiMe$_2$Ph to SiMePh$_2$ or SiEt$_3$ resulted in only slightly decreased yield, but the SiPh$_3$-substituted reagent **1d** afforded much lower yield (Table 1, entries 3–5). It was found that Mn

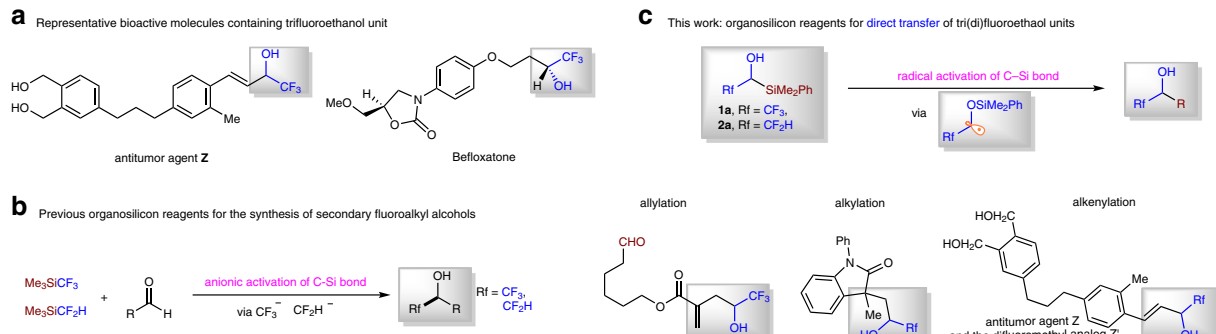

**Fig. 1 Secondary fluoroalkyl alcohol synthesis with fluorinated organosilicon reagents. a** Representative bioactive molecules containing trifluoroethanol unit. **b** Organosilicon reagents has been used in the synthesis for secondary fluoroalkyl alcohols via α-fluoro carbanions transfer. **c** Organosilicon reagents for direct transfer of tri(di)fluoroethanol units via radical activation strategy is developed (this work).

**Fig. 2 Preparation of fluoroalkyl group transfer reagents 1a and 2a.** With commercially available **3**, reagents **1a** and **2a** are easily prepared in three steps.

**Fig. 3 Attempts for anionic activation and design of radical activation.**
**a** Attempts of anionic activation for trifluoroethanol transfer failed. **b** Facile β-F anion elimination might be the reason of the failure of anionic activation strategy; we designed a radical activation strategy based on the proposal that it is difficult to eliminate high-energy fluorine radical.

(OAc)$_2$·4H$_2$O can also be used as catalyst (Table 1, entry 6). When 20 mol% of Mn(OAc)$_2$·4H$_2$O was used, **1a/7a**/TBPB is 1/2/2.5, a yield of 81% can be detected by $^{19}$F NMR with PhCF$_3$ as an internal standard (Table 1, entry 7). Control experiments verified that both Mn(II) catalyst and oxidant are necessary for the success of the reaction (Table 1, entries 8 and 9). When 2,2,2-trifluoroethanol was used instead of compound **1a** in our reaction, no conversion of trifluoroethanol was observed (Table 1, entry 10). The high BDE of C–H bond (409 kJ/mol) in trifluoroethanol might be one of the reasons for the difficulty in the generation of desired radical directly from trifluoroethanol under mild conditions[26–29] (There are limited reports on the generation of carbon radicals from trifluoroethanol under harsh conditions, see refs. [27–29]). The attempt to use Smith's condition[19] to achieve the reaction between **1a** and **7a** failed to afford any amount of product **8a** or **9a**, highlighting the influence of CF$_3$ group on the reactivity of reagent **1a**.

**Synthesis of α-trifluoromethylated homoallylic alcohols**. We next explored the scope of the allylic substitution reaction (Fig. 4). Both Mn(OAc)$_3$·2H$_2$O and Mn(OAc)$_2$·4H$_2$O are efficient catalysts for the transformation. When TBAF was used to quench the reaction, the silyl ether was converted to alcohol in one pot, and compound **9a** was isolated in 62% yield. It was found that a variety of allylic sulfones[30–34] bearing different groups could be

**Table 1 Reaction optimization[a].**

| Entry | 1a/7a/oxidant | Catalyst | Oxidant | t (h) | Yield (%) |
|---|---|---|---|---|---|
| 1 | 1/1.2/0 | Mn(OAc)$_3$·2H$_2$O (2 equiv.) | Without oxidant | 18 | 68 |
| 2 | 1/1.5/2.5 | Mn(OAc)$_3$·2H$_2$O (20 mol%) | TBPB | 12 | 61 |
| 3[b] | 1/1.5/2.5 | Mn(OAc)$_3$·2H$_2$O (20 mol%) | TBPB | 12 | 59 |
| 4[c] | 1/1.5/2.5 | Mn(OAc)$_3$·2H$_2$O (20 mol%) | TBPB | 12 | 58 |
| 5[d] | 1/1.5/2.5 | Mn(OAc)$_3$·2H$_2$O (20 mol%) | TBPB | 12 | 45 |
| 6 | 1/1.5/2.5 | Mn(OAc)$_3$·2H$_2$O (20 mol%) | TBPB | 18 | 62 |
| 7 | 1/2/2.5 | Mn(OAc)$_2$·4H$_2$O (20 mol%) | TBPB | 18 | 81 |
| 8 | 1/2/0 | Mn(OAc)$_2$·4H$_2$O (20 mol%) | Without oxidant | 18 | 0 |
| 9 | 1/2/2.5 | Without catalyst | TBPB | 18 | 0 |
| 10[e] | 1/2/2.5 | Mn(OAc)$_2$·4H$_2$O (20 mol%) | TBPB | 18 | 0 |

[a]**1a** was used as the reagent, otherwise noted; the yield of the product **8** was determined by $^{19}$F NMR with PhCF$_3$ as an internal standard.
[b]**1b** was used instead of **1a**.
[c]**1c** was used instead of **1a**.
[d]**1d** was used instead of **1a**.
[e]Trifluoroethanol was used instead of **1a**.

1a, [Si] = SiMe$_2$Ph;
1b, [Si] = SiPh$_2$Me;
1c, [Si] = SiEt$_3$;
1d, [Si] = SiPh$_3$,

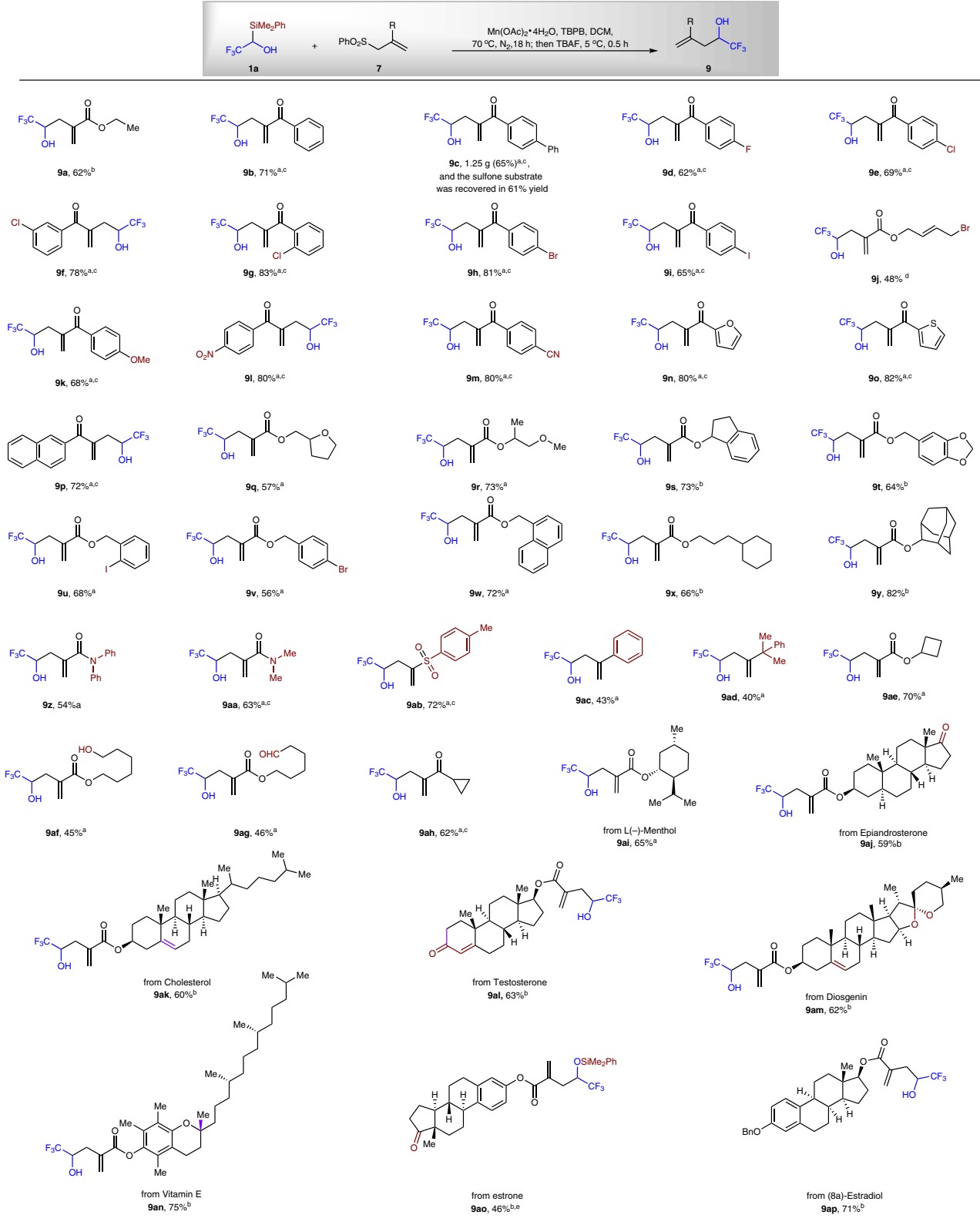

**Fig. 4 Scope for the allylation of reagent 1a.** [a]1a/**7** = 1/3, [b]**1a**/**7** = 1/2, and [c]Mn(OAc)$_3$·2H$_2$O was used instead of Mn(OAc)$_2$·4H$_2$O. [d]2 equivalent of Mn(OAc)$_3$·2H$_2$O was used without TBPB. [e]Instead of TBAF, water was used to quench the reaction.

used as the radical acceptors, affording the α-CF$_3$-substituted homoallylic alcohols **9a–9ap** in 40–83% yields. This reaction can be scaled up, and a yield of 65% was obtained for compound **9c** (1.25 g isolated), and the sulfone **7c** was recovered in 61% yield

after silica gel chromatography. It is worth noting that the current allylation protocol can tolerate many functional groups. The Csp$^2$–F, Csp$^2$–Cl, Csp$^2$–Br, Csp$^2$–I bond can be kept after the reaction, affording compounds **9d–9i**, **9u**, and **9v** in 56–83%

yields. Moreover, allylic bromide can also be tolerated (**9j**, 48% yield). These functional groups are well-known convertible motifs under transition metal-mediated/catalyzed reactions. Electron donating OMe group, electron withdrawing groups, such as NO₂, CN can also be tolerated, affording compounds **9k–9m** in 68–80% yields. Electron-rich heterocyclic groups, such as furyl and thienyl are also be tolerated (**9n**, 80% yield; **9o**, 82% yield). Naphthyl-substituted product was also successfully made via the Mn-catalyzed substitution reaction, affording alcohol **9p** in 72% yield. A variety of alkyl ethers and benzylic alcohol-derived esters are tolerated under the current oxidation conditions, and alcohols **9q–9w** were obtained in 56–73% yields. Not only ester-substituted and ketone-substituted allylic sulfones, but also amide, sulfone, phenyl, and alkyl groups-substituted allylic sulfones can be applied in the current substitution process, affording corresponding α-CF₃-substituted homoallylic alcohols (**9z–9ad**, 40–72% yields). Moreover, this substitution process is compatible with many base-sensitive functionality, such as primary alcohol (**9af**, 45% yield), alkyl aldehyde (**9ag**, 46% yield), alkyl ketone (**9ah**, 62% yield; **9aj**, 59% yield; **9al**, 63% yield; **9ao**, 46% yield). It is worthy to note that the above-mentioned aldehyde and ketone-containing products are challenging to be synthesized by methods based on these functional groups[3]. In addition, the current radical allylic substitution can be applied in the functionalization of complex molecules, such as the derivatives from biologically active naturally occuring epiandrosterone, cholesterol, testosterone, diosgenin, vitamin E, estrone, and (8α)-estradiol, affording corresponding alcohols **9ai–9ap** in 46–75% yields.

**Synthesis of α-trifluoromethylated alkyl alcohols.** After achieving the radical C–Si bond activation to access α-trifluoromethylated homoallylic alcohols, we wondered whether the same strategy can be applied in double functionalization of alkenes to prepare α-trifluoromethylated alkyl alcohols. Acryl amides **10** were chosen to test the possibility[35–38]. To our delight, under Mn(II)/TBPB conditions, various acryl amides can be converted to corresponding trifluoromethylated alcohols **11** in 63–91% yield (Fig. 5). Me, Ph, and Bn groups on the N of amides do not affect the reaction. The reaction tolerates halides, such as F

and Cl. Both electron-donating OMe and electron-withdrawing CO₂Me on the arenes were maintained after the reaction. It is worthy to note that compounds **11** are difficult to be synthesized through the nucleophilic trifluoromethylation reaction of aldehydes with TMSCF₃, because the aldehydes themselves need multistep synthesis[10].

**Synthesis of α-trifluoromethylated allylic alcohols.** Allylic alcohols are important synthetic intermediates for a variety of transformations. Moreover, fluorinated compounds possess unique chemical, physical, and biological properties. Therefore, the development of one step synthesis of α-trifluoromethylated allylic alcohols from readily available starting materials are highly desired. Previous methods to prepare α-trifluoromethylated allylic alcohols mainly rely on nucleophilic trifluoromethylation of α,β-unsaturated aldehydes[39–43]. α,β-Unsaturated carboxylic acids are easy to be synthesized and many of them are commercially available. Moreover, they are more stable under atmosphere than corresponding α,β-unsaturated aldehydes. They have been used in the decarboxylative trifluoromethylation reactions for the synthesis of CF₃-substituted alkenes[44,45]. Encouraged by the success of synthesizing homoallylic and alkyl alcohols with **1a** as reagent under the radical C–Si bond activation conditions, we studied the reaction between **1a** and acids **12** to prepare α-trifluoromethylated allylic alcohols (Fig. 6). The reaction conditions are slightly different from the above allylation and alkylation reactions: (1) both hexane and DCM could be used as solvent; (2) the desilylation step was performed under lower temperature (−10 vs. 5 °C) to avoid side reactions. It can be found from Fig. 7 that various α,β-unsaturated carboxylic acids containing F, Cl, Br, MeO, BnO, and CF₃ substituents were successfully converted to corresponding α-trifluoromethylated allylic alcohols in 51–75% yield. To the best of our knowledge, there has been no report on the synthesis of α-trifluoromethylated allylic alcohols with unsaturated carboxylic acids before this work.

**Radical C–Si activation for difluoroethanol unit transfer.** Difluoromethyl group (CF₂H) is isopolar and isosteric to an SH

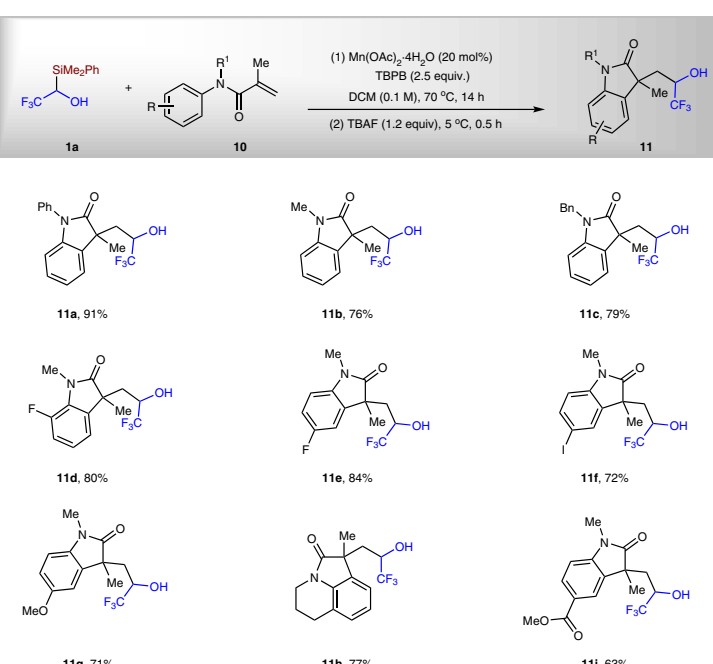

**Fig. 5 Synthesis of α-trifluoromethylated alkyl alcohols.** All reactions were run under the standard conditions with **1a/10** = 1/2.

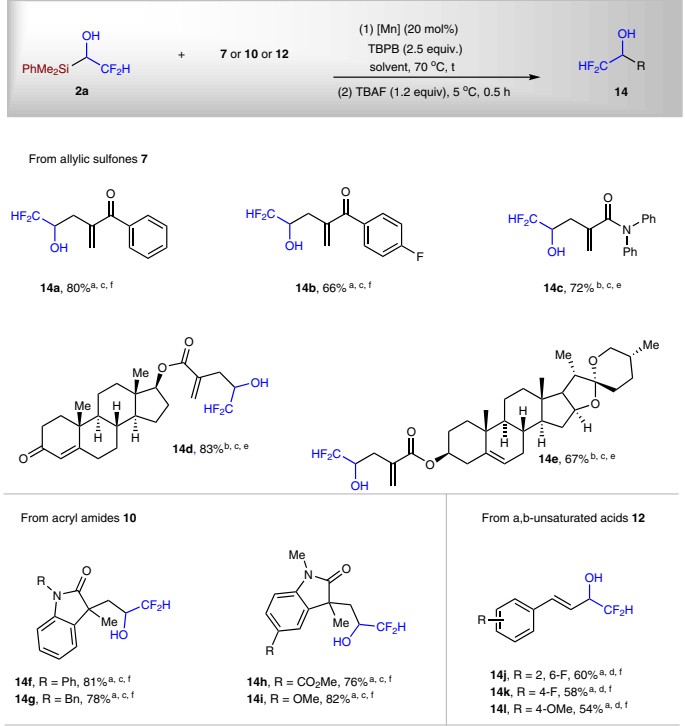

**Fig. 6 Synthesis of α-trifluoromethylated allylic alcohols.** [a]Reactions were run under standard conditions with **1a**/**12** = 1/2. [b]DCM was used as solvent instead of hexanes.

**Fig. 7 Synthesis of α-difluoromethylated alcohols.** [a]Mn(OAc)$_3$·2H$_2$O (20 mol%) was used as catalyst. [b]Mn(OAc)$_2$·4H$_2$O (20 mol%) was used as catalyst. [c]DCM (0.1 M) was used as solvent. [d]Hexane (0.4 M) was used as solvent. [e]$t$ = 18 h and [f]$t$ = 14 h.

or OH group, and can also behave as a hydrogen donor through hydrogen bonding[46,47]. Therefore, difluoromethylated compounds are strong candidate for drugs. There are examples showing that the CF$_2$H-containing compounds exhibit higher bioactivity than their CF$_3$-containing counterparts[48,49]. Encouraged by the above success of radical C–Si bond activation to access trifluoromethylated allylic, alkyl, and alkenyl alcohols, we extended the strategy to directly transfer 2,2-difluoroethanol to organic molecules with reagent **2a** (Fig. 7). To the best of our knowledge, there has been no report of the synthetic application

of carbon radical from 2,2-difluoroethanol. Under the similar reaction conditions as that of **1a**, with allylic sulfones as substrates, difluoromethylated homoallylic alcohols **14a**–**14c** were prepared in 66–80% yield. Late-stage functionalization of two complex molecules are also successful; **14d** and **14e** were isolated in 83% and 67% yield, respectively. The reactions of acryl amides performed well, affording **14f**–**14i** in 76–82% yield. Ph, Bn, and Me groups on the N atom were tolerated; both electron-withdrawing CO$_2$Me and electron-donating OMe groups on the aromatic ring were maintained after the reactions. Three

**Fig. 8 Synthesis of antitumor agent Z and its difluoromethyl analog Z′.** Tf₂O trifluoromethanesulfonic anhydride, DPPP bis(diphenylphosphino)propane, TFA trifluoroacetic acid, DIBAL-H diisobutylaluminum hydride.

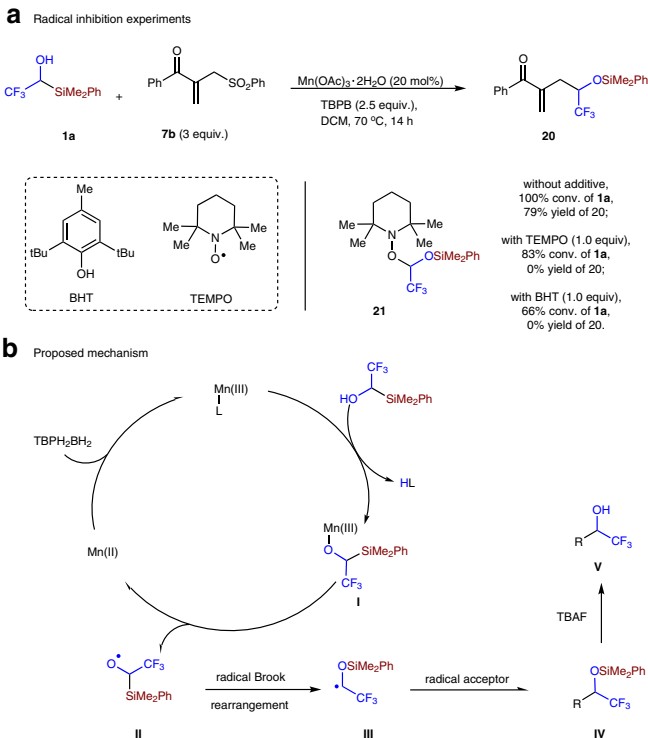

**Fig. 9 Radical inhibition experiments and proposed mechanism. a** TEMPO and BHT efficiently inhibited the allylation reaction, supporting radical process might be involved. **b** A plausible mechanism is proposed.

examples of α,β-unsaturated carboxylic acids demonstrated the utility of synthesis of difluoromethyl group-substituted allylic alcohols **14j–14l** in synthetically useful yield.

**Synthesis of antitumor agent Z and its difluoromethyl analog Z′.** After achieving the direct transfer of 2,2,2-trifluoroethanol and 2,2-difluoroethanol to simple α,β-unsaturated carboxylic acids, we tested whether our methodology can be applied in the synthesis of antitumor agent **Z** (Fig. 8)[5]. Starting from known compound **15**[5], the protection of phenol afforded compound **16** in 97% yield. α,β-Unsaturated carboxylic acid **17** was then synthesized in 76% yield after Heck reaction and selective hydrolysis of the ester. The radical reactions performed well under our standard conditions, affording compounds **18** and **19** in 71% yield and 69% yield, respectively. Hydrolysis of the esters afforded the final product **Z** and **Z′** in 94% and 89% yield, respectively.

**Mechanism of the study**. The radical inhibition experiments revealed that addition of 1 equiv. of TEMPO or BHT can completely inhibit the allylation reaction, albeit more than 60% of compound **3a** were consumed in both cases (Fig. 9a). In addition, compound **21** was detected by HRMS when TEMPO was added into the reaction (for details, see Supporting Information). Therefore, radical process might be involved in current substitution reaction. Our investigation revealed that Mn(OAc)₃·2H₂O is able to mediate the reaction without external oxidant, but Mn(OAc)₂·4H₂O cannot mediate the reaction without TBPB (Table 1, entries 1 and 8). Based on these results, we propose a possible mechanism as shown in Fig. 9b. Ligand exchange between Mn(III) species and alcohol **1a** might generate intermediate **I**, which undergoes homolysis to produce alkoxyl radical **II** and Mn(II) intermediate[50–53]. Carbon radical **III** would be generated through Brook rearrangement, and then undergo further reaction to generate product **IV**. Mn(III) catalyst is likely to be regenerated by the oxidation of Mn(II) by TBPB. The alcohol product **V** would be generated after the desilylation step. It is worthy to note that the reaction pathways in the allylation, alkylation, and alkenylation reactions are probably different (for the proposed possibilities, see Supplementary Figs. 13–15). As for the nature of radical **III**, we propose that they are nucleophilic radicals. The following facts support this proposal: (1) the reaction partners in the above three kinds of reactions are electron-deficient olefins; (2) more electron-deficient substrate afforded higher yield (for example, Fig. 5, **9ac**, 43% yield; **9ae**, 70% yield); (3) the carbon radical generated from trifluoroethanol under γ-ray irradiation could add to highly electrophilic hexafluoro-2-butyne[27].

**Discussion**

In conclusion, we have developed two fluorinated organosilicon reagents, which were used in direct transfer of trifluoroethanol and difluoroethanol units into organic molecules. A radical C–Si bond activation strategy was developed to solve the problem of β-fluorine elimination in anionic activation methods. Upon intramolecular activation of C–Si bond by alkoxyl radicals, the β-fluoro carbon radicals were generated and participated in efficient allylation, alkylation, and alkenylation reactions, enabling efficient synthesis of numerous fluoroalkyl alcohols. The broad applicability and general utility of the approach are highlighted by late-stage introduction of fluoroalkyl groups to complex molecules and the synthesis of antitumor agent **Z** and its difluoromethyl analog **Z′**. Further application of the radical C–Si bond activation of organosilicon reagents are underway in our lab.

**Methods**

**Typical synthesis of compound 9**. Under N₂ atmosphere, to a dried 10 mL Schlenk tube equipped with a magnetic stir bar containing Mn(OAc)₂·4H₂O

(14.7 mg, 0.06 mmol, 20 mol%) was added DCM (3 mL, 0.1 M), **1a** (70.2 mg, 0.3 mmol), **7a** (152.4 mg, 0.6 mmol, 2.0 equiv.), and TBPB (145.7 mg, 0.75 mmol, 2.5 equiv.) sequentially. The tube was sealed, and the resulting mixture was kept stirring at 70 °C in a heating block for 18 h. The mixture was then cooled to 5 °C with ice bath, TBAF (1.0 M in THF, 0.36 mL, 0.36 mmol, 1.2 equiv.) was added and the resulting mixture was stirred at 5 °C for 0.5 h. The reaction mixture was quenched with water (2 mL), extracted with DCM (3 × 10 mL) and organic phase was combined and washed with brine, dried over Na$_2$SO$_4$, concentrated under reduced pressure. The crude product was purified with column chromatography on silica gel (200–300 mesh) and PE/EA (20/1–10/1, v/v) as eluent to afford 40.0 mg of compound **9a** as a colorless oil (62% yield).

## Data availability
The authors declare that all data supporting the findings of this study are available within the article and Supplementary Information files, and also are available from the corresponding author on reasonable request.

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

## Acknowledgements

We are grateful to NSFC (21901191), Fundamental Research Funds for the Central Universities (2042018kf0023, 2042019kf0006), State Key Laboratory of Bioorganic & Natural Products Chemistry (BNPC18237) and Wuhan University for financial support. We thank Prof. Tobias Ritter (Max-Planck-Institut für Kohlenforschung, Germany), Prof. Jinbo Hu (Shanghai Institute of Organic Chemistry, China), and Prof. Amir H. Hoveyda (Boston College, USA) for helpful discussion. We are thankful to Prof. Aiwen Lei and Prof. Xumu Zhang at Wuhan University for the generous provision of laboratory and facilities.

## Author contributions

X.S. designed and directed the investigations and composed the manuscript with revisions provided by the other authors. X.C., X.G., and X.S. were involved in the invention of the reagents **1a** and **2a**. X.C. and X.G. developed the catalytic method. X.C., X.G., Z.L., G.Z., and Z.Z. studied the substrate scope. X.C., X.G., Z.L., G.Z., Z.Z., W.Z., S.L., and X.S. were involved in the analyses of results and discussions of the project.

## Competing interests

The authors declare the following competing interests: X.S., X.C., and X.G. have applied a patent based on the work of this manuscript. All other authors declare no competing interests.
