## [Peer Review File · Nature Communications]

Reviewers' comments:

Reviewer #1 (Remarks to the Author):

The authors present an oxidative radical Brook rearrangement reaction using the novel transfer reagents 1 and 2 mediated by substoichiometric Mn(OAc)₃ and a stoichiometric peroxide. Control experiments depicted in Table 1 established that both are required for the reaction to occur.

The method is obviously aimed at synthesis chemists who wish to introduce fluorinated motifs into compounds but it must be stated that the silicon reagents are not that easy to prepare. Compound 1 requires 5 steps (or operations) - however the steps are counted this is not straightforward chemistry and the requirement for HMPA will deter many people (especially those in industry). In future work the authors could potentially develop a more process friendly route to these reagents if they wish the methodology to be used outside their laboratory.

Above comments notwithstanding the authors have demonstrated an impressive range of substrates and prepared antitumor agent Z and an analog.

The proposed mechanism section is a little vague but I appreciate that the manuscript is a communication. The presence of radicals is very likely and inferred from the TEMPO/BHT experiments. One aspect that has been overlooked in Fig. 11 is the source of H following the addition of the radical III to the radical acceptor. This generates another radical that requires H atom quench.

I think that this work is very good but I feel for publication in this journal the methodology needs a high degree of novelty and the radical oxidative Brook rearrangement that is at the core of this process has already been reported by Amos Smith (J. Am. Chem. Soc. 139, 9487-9490 (2017) - reference 19 in the manuscript. The authors have built upon this work with an alternative silicon reagent and oxidant system. For this reason I think that the work is great but it should be published in specialist journal.

Reviewer #2 (Remarks to the Author):

The manuscript by Shen and co-workers reports two new fluorinated organosilicon reagents which were used in direct transfer of trifluoroethanol and difluoroethanol units into organic molecules.

Unlike the previously known C-Si activation mode (ionic cleavage to generate a anionic R_f species), the authors developed a radical C-Si bond activation strategy to solve the problem of β-fluorine elimination in anionic activation methods. Upon intramolecular activation of C-Si bond by alkoxy radicals, the β-fluoro carbon radicals were generated and participated in efficient allylation, alkylation and alkenylation reactions, enabling efficient synthesis of numerous fluoroalkyl alcohols. The broad applicability and general utility of the approach are highlighted by late-stage introduction of fluoroalkyl groups to complex molecules and the synthesis of antitumor agent Z and its difluoromethyl analog Z'. I am pleased to recommend the publication of this nice article in Nature Communications.

Respond to the reviewer's comments

Reviewers' comments:

Reviewer #1 (Remarks to the Author):

The authors present an oxidative radical Brook rearrangement reaction using the novel transfer reagents **1** and **2** mediated by substoichiometric $\text{Mn}(\text{OAc})_3$ and a stoichiometric peroxide. Control experiments depicted in Table 1 established that both are required for the reaction to occur.

Thanks for the comments. We want to clarify that when $\text{Mn}(\text{OAc})_3$ was used as agent, the reaction can occur without peroxide (Table 1, entry 1).

The method is obviously aimed at synthesis chemists who wish to introduce fluorinated motifs into compounds but it must be stated that the silicon reagents are not that easy to prepare. Compound **1** requires 5 steps (or operations) - however the steps are counted this is not straightforward chemistry and the requirement for HMPA will deter many people (especially those in industry). In future work the authors could potentially develop a more process friendly route to these reagents if they wish the methodology to be used outside their laboratory.

We regret that we did not succeed in convincing Reviewer #1 of the easy synthesis of our reagents. But we believe that our synthesis routes are 3-step reactions not 5-step (or operation) reactions. It is not fair to count "allowing the temperature to rt" and "adding a compound into the flask" as two steps. Three-step synthesis is not difficult for synthetic chemists, and undergraduate students in my lab successfully performed these reactions.

*Actually, there is 2-step synthesis of reagent **1d** without HMPA. A 76% yield was reported for trifluoroacetyltriphenylsilane by one step reaction with silyl lithium and trifluoroacetic anhydride (ref 25 in the original submitted manuscript, *Tetrahedron Lett.* **1992**, 33, 1221-1224). Reduction of trifluoroacetyltriphenylsilane under our reaction condition would afford **1d**. We added one sentence in the revised manuscript to refer to this paper (ref 24 in the revised manuscript). Moreover, the synthesis of Et_3Si substituted reagent **1c** in our 3-step synthesis does not need HMPA and the details were provided in Supplementary Information. Reviewer #1 might missed these information.*

*In addition, the silyl group does not affect the efficiency of our radical reaction that much and the SiPh_3 , SiEt_3 , SiPh_2Me and SiMe_2Ph substituted reagents afforded desired products in 45~61% yield before further optimization (Table 1, entries 2-5). People who worry about HMPA could choose to use reagents **1c** or **1d** to optimize conditions to get higher yield.*

Above comments notwithstanding the authors have demonstrated an impressive range of substrates and prepared antitumor agent **Z** and an analog.

Thanks to Reviewer #1 for pointing out our contribution.

The proposed mechanism section is a little vague but I appreciate that the manuscript is a communication. The presence of radicals is very likely and inferred from the TEMPO/BHT experiments. One aspect that has been overlooked in Fig. 11 is the source of H following the

addition of the radical III to the radical acceptor. This generates another radical that requires H atom quench.

Thanks to Reviewer #1 for the support of our proposed radical mechanism. Allylic sulfones (ref 34), acryl amides (refs 36-37) and α,β -unsaturated carboxylic acids (ref 46) were reported in radical reactions. Since we report three different reactions in this manuscript (allylation with allylic sulfones, alkylation with acryl amides and alkenylation with α,β -unsaturated carboxylic acids), we decided to propose a unified mechanism. **However, not all of these reactions require H atom to quench radicals.** Please see the proposed mechanism below.

Fig. R-1. Proposed Mechanism for allylation via radical C-Si activation

The radical inhibition experiments (Fig 10) indicate that a radical process might be involved. We found that $\text{Mn}(\text{OAc})_3 \cdot 2\text{H}_2\text{O}$ is able to mediate the reaction without external oxidant, but $\text{Mn}(\text{OAc})_2 \cdot 4\text{H}_2\text{O}$ can not mediate the reaction without TBPB (Table 1, entries 1 and 8 in the manuscript). The HRMS analysis of the reaction mixture of 1a and 7a suggests the generation of benzenesulfonyl benzoic anhydride, tert-butyl benzenesulfonate, benzenesulfonic acid and benzenesulfinic acid as by-products. Based on these experimental results and literature about allylation from allylic sulfone (ref 34 in the manuscript), we propose a possible mechanism (Fig. R-1). Ligand exchange between Mn(III) species and alcohol 1a might generate intermediate I, which undergoes homolysis to produce alkoxy radical II and Mn(II) intermediate. Carbon radical III would be generated through Brook rearrangement, and then undergo radical addition reaction to generate intermediate IV. Compound V would be generated after β -elimination of sulfonyl radical. The alcohol product TM would be generated after the desilylation step. Mn(III) catalyst is likely to be regenerated by the oxidation of Mn(II) by TBPB.

The sulfonyl radical could be transformed to sulfinic acid via H atom abstraction reaction under the reaction conditions, which might be supported by the HRMS data. But there are other possible pathways to consume the sulfonyl radicals, based on the HRMS data. The sulfonyl radical is likely to be captured by TBPB, generating the side-product benzenesulfonyl benzoic anhydride. The sulfonyl radical might also be oxidized and captured by PhCO_2^- to generate benzenesulfonyl benzoic anhydride. Meanwhile, sulfonyl radical could react with TBPB or tert-butoxy radical to form tert-butyl benzenesulfonate. Benzenesulfonyl benzoic anhydride and tert-butyl benzenesulfonate could be hydrolyzed to generate benzenesulfonic acid.

Benzenesulfonyl benzoic anhydride: HRMS (ESI, m/z): calcd for $C_{13}H_{10}NaO_4S^+$ (M+Na) $^+$: 285.0192; Found: 285.0195.

tert-Butyl benzenesulfonate: HRMS (ESI, m/z): calcd for $C_{10}H_{14}NaO_3S^+$ (M+Na) $^+$: 237.0556; Found: 237.0561

Benzenesulfinic acid: HRMS (ESI, m/z): calcd for $C_6H_6NaO_2S^+$ (M+Na) $^+$: 164.9981; Found: 164.9975.

Benzenesulfonic acid: HRMS (ESI, m/z): calcd for $C_6H_7O_3S^+$ (M+H) $^+$: 159.0110; Found: 159.0115.

For the alkylation reaction, we propose that radical III would be generated following similar mechanism as that in the allylation reaction (Fig. R-2). When an acryl amide was used as the radical acceptor instead of an allylic sulfone, we propose that radical III could undergo addition reaction to generate intermediate IV', which undergo intramolecular addition to generate intermediate V'. Aromatization reaction via radical oxidation and deprotonation would generate compound VI'. The alcohol product TM' would be generated after the desilylation step. Mn(III) catalyst is likely to be regenerated by the oxidation of Mn(II) by TBPB. Similar oxidative aromatization process was also proposed in the Fe and Ag catalyzed radical reactions of acryl amides (refs 36 and 37 in the manuscript).

We do not prefer H atom quench of radicals in our alkylation reactions.

Fig. R-2. Proposed Mechanism for alkylation via radical C-Si activation

There are reports on radical decarboxylative alkenylation with α,β -unsaturated carboxylic acids (ref 46 in this manuscript and Chem. Sci. 3, 2853-2858 (2012)). Based on our experimental results and literature reports, we propose a possible mechanism for our reaction as shown in Fig. R-3. Ligand exchange between Mn(III) species and alcohol 1a might generate intermediate I, which undergoes homolysis to produce alkoxy radical II and Mn(II) intermediate. Carbon radical III would be generated through Brook rearrangement, and then undergo radical addition reaction via two possible pathways to generate TM''.

Pathway a: addition of radical III to the α -position of the double bond in an α,β -unsaturated carboxylic acid would generate intermediate IV'''. Intermediate IV''' was oxidized to cation

intermediate V'' which then eliminated carbon dioxide and proton to generate the product VI'' . Similar proposal was proposed in Ni-catalyzed radical *alkenylation with α,β -unsaturated carboxylic acids* (ref 46 in this manuscript). The alcohol product TM'' would be generated after the desilylation step.

Pathway b: compound A could be transformed to compound B via ligand exchange process. Addition of radical III to the α -position of the double bond in compound B would generate intermediate IV'' , which then eliminated carbon dioxide and Mn(II) to generate compound VI'' . The alcohol product TM'' would be generated after the desilylation step. Similar proposal was proposed in Cu-catalyzed radical *alkenylation with α,β -unsaturated carboxylic acids* (Chem. Sci. 3, 2853-2858 (2012)). The alcohol product TM'' would be generated after the desilylation step.

Mn(III) catalyst is likely to be regenerated by the oxidation of Mn(II) by TBPB.

We do not prefer H atom quench of radicals in our alkenylation reactions.

Fig. R-3 Proposed Mechanism for alkenylation via radical C-Si activation

The mechanism proposal for three type of reactions (allylation, alkylation and alkenylation) was added in the revised supplementary Information.

I think that this work is very good but I feel for publication in this journal the methodology needs a high degree of novelty and the radical oxidative Brook rearrangement that is at the core of this process has already been reported by Amos Smith (J. Am. Chem. Soc. 139, 9487-9490 (2017) - reference 19 in the manuscript). The authors have built upon this work with an

alternative silicon reagent and oxidant system. For this reason I think that the work is great but it should be published in specialist journal.

Thanks to Reviewer #1 for agreeing that our work is very good. We regret that we did not succeed in convincing Reviewer 1 of the novelty of our chemistry. As Reviewer 1 pointed out we have developed an alternative silicon reagent and oxidant system. In my opinion, this is indeed one of the novelty in our chemistry. It's worthy to note that Prof. Smith's work advanced the development of radical Brook rearrangement by developing alternative reaction conditions, because the oxidative radical Brook rearrangement reported early in 2000 possesses limited synthetic application (ref 18 in our manuscript). However, Smith's conditions failed in our reactions and no product was observed (shown below). Full conversion of 1a probably resulted from the anion Brook rearrangement-fluorine elimination process. A base such as KOPiv is required in Smith's chemistry, but the addition of the base is detrimental to our chemistry. This result recalls me another novelty of our chemistry: using radical strategy to solve the fluorine chemistry problem of in anion strategy (Figure 4 in the manuscript). The fluorine chemistry is different from non-fluorine chemistry. Besides, we would be grateful if Reviewer #1 could understand another novelty of our chemistry: we developed new organosilicon reagents to directly transfer important structural motifs into molecules. These compounds are troublesome to synthesize by other methods.

We added one sentence to report the failure of Smith's condition in our reaction in the revised manuscript.

Reviewer #2 (Remarks to the Author):

The manuscript by Shen and co-workers reports two new fluorinated organosilicon reagents which were used in direct transfer of trifluoroethanol and difluoroethanol units into organic molecules. Unlike the previously known C-Si activation mode (ionic cleavage to generate a anionic Rf species), the authors developed a radical C-Si bond activation strategy to solve the problem of β -fluorine elimination in anionic activation methods. Upon intramolecular activation of C-Si bond by alkoxy radicals, the β -fluoro carbon radicals were generated and participated in efficient allylation, alkylation and alkenylation reactions, enabling efficient synthesis of numerous fluoroalkyl alcohols. The broad applicability and general utility of the approach are highlighted by late-stage introduction of fluoroalkyl groups to complex molecules and the synthesis of antitumor agent Z and its difluoromethyl analog Z'. I am pleased to recommend the publication of this nice article in Nature Communications.

Thanks to Reviewer #2 for the comments.

Reviewers' comments:

Reviewer #1 (Remarks to the Author):

The authors have provided clear replies to the questions that I raised in my review and have made some good improvements to the manuscript.

Regarding the reagents and their synthesis the author's reply is well made but I will just state this. I run multiple collaborative projects with the pharmaceutical and agrochemical industries and I can say with confidence that their reagents are very unlikely to be synthesised industry (at least by the current methods). If the authors want people to use these reagents they need to be commercialized. This does not affect my judgement of the paper but I mention it because academic research groups don't really use fluorination processes of this type. This method is ideally suited for drug discovery projects.

I think that the addition of the extra mechanistic schemes to the SI will be helpful for readers.

I still have some reservations regarding the novelty of the manuscript because of the work that has previously been reported by Amos Smith and co-workers.

Reviewer #3 (Remarks to the Author):

The manuscript "Direct Transfer of Tri(di)fluoroethanol Units Enabled ..." by Shen and coworkers describes an interesting use of new silicon reagents which allow the straightforward introduction of trifluoroethanol or difluoroethanol units on diverse scaffolds. More precisely a radical intermediate is generated in oxidative conditions which then can react through radical allylation or intermolecular additions. The generation of the radical intermediate takes place through a radical Brook rearrangement. This could be reminiscent of recent work by Amos Smith but the set of silicon reagents is different and the conditions too. This revised version answers clearly to previous remarks and can be accepted provided it also answers to the following points:

- Fig. 1, correct "antitumer"
- The BDE values which are mentioned on page 3 should be given.
- What is missing in this manuscript is some bibliographical data about the trifluoroethanol and difluoroethanol radicals. Have they been generated and used ? They look like nucleophilic radicals. It would be nice to have some calculations to show that.
- Why in some cases Mn(II)/TBHP conditions are used and in other cases Mn(III)/TBPB conditions, for instance Fig. 6 vs Fig. 7 ? or also in Fig. 8 ?
- On page 7, in the conclusion, the sentence "it is worthy to note that the reaction pathways of radical ..." should be changed. As rationalized in the SI, the three pathways have different mechanistic manifolds.

Response to the reviewers' comments

Reviewers' comments:

Reviewer #1 (Remarks to the Author):

The authors have provided clear replies to the questions that I raised in my review and have made some good improvements to the manuscript.

Response: Thanks to reviewer 1 for the positive comments.

Regarding the reagents and their synthesis the author's reply is well made but I will just state this. I run multiple collaborative projects with the pharmaceutical and agrochemical industries and I can say with confidence that their reagents are very unlikely to be synthesised industry (at least by the current methods). If the authors want people to use these reagents they need to be commercialized. This does not affect my judgement of the paper but I mention it because academic research groups don't really use fluorination processes of this type. This method is ideally suited for drug discovery projects.

Response: Thanks to reviewer 1 for the positive comments: "this method is ideally suited for drug discovery projects".

Actually, two companies are interested in selling these reagents after talking with us. We will make these reagents commercially available. Besides that, we have submitted a patent application about these reagents and the methodology which will be disclosed in this manuscript. We added this information in "competing interests" part in the revised manuscript. The commercialization of reagents will promote more academic research groups to use our methodology. We appreciate reviewer 1 for the suggestion to commercialize our reagents.

I think that the addition of the extra mechanistic schemes to the SI will be helpful for readers.

Response: Thanks to reviewer 1 for the positive comments.

I still have some reservations regarding the novelty of the manuscript because of the work that has previously been reported by Amos Smith and co-workers.

Response: We regret that we did not fully succeed in convincing Reviewer 1 of the novelty of our manuscript. We know Prof. Amos Smith's contribution in the application of radical Brook rearrangement in organic synthesis by introducing different condition with known organosilicon reagents. Our contribution here is introducing novel reagents and conditions. Prof. Smith's chemistry failed in the reaction with our reagents. Moreover, β -fluorine anion elimination is a known problem in fluorine chemistry (refs 24-26 in the manuscript), and here we provide a way to solve the problem (Figure 4 in the manuscript). Last but not least, tri(di)fluoroethanol units are important, and we disclose a type of novel reagents and a one-step method to directly incorporate these units into molecules, including complex bioactive molecules.

Reviewer #3 (Remarks to the Author):

The manuscript "Direct Transfer of Tri(di)fluoroethanol Units Enabled ..." by Shen and coworkers describes an interesting use of new silicon reagents which allow the straightforward introduction of trifluoroethanol or difluoroethanol units on diverse scaffolds. More precisely a radical intermediate is generated in oxidative conditions which then can react through radical allylation or intermolecular additions. The generation of the radical intermediate takes place through a radical Brook rearrangement. This could be reminiscent of recent work by Amos Smith but the set of silicon reagents is different and the conditions too. This revised version answers clearly to previous remarks and can be accepted provided it also answers to the following points:

Response: Thanks to Reviewer 3 for pointing out the novelty and the difference between our work and Prof. Smith's work.

- Fig. 1, correct "antitumer"

Response: We are sorry for the typo. We have corrected "antitumer" in fig 1 to "antitumor"

- The BDE values which are mentioned on page 3 should be given.

Response: Thanks for the suggestion. The BDE of the C-H bond of 2,2,2-trifluoroethanol reported in ref. 28 was added in the revised manuscript. We are sorry that we do not have the exact BDE data of C-Si bonds of our reagents. We revised the sentence as below:

"The higher BDE of C-H bond in trifluoroethanol than that of C-Si bond in compound 1a might be one of the reasons ..." was changed to "The high BDE of C-H bond (409 KJ/mol) in trifluoroethanol might be one of the reasons ..."

- What is missing in this manuscript is some bibliographical data about the trifluoroethanol and difluoroethanol radicals. Have they been generated and used? They look like nucleophilic radicals. It would be nice to have some calculations to show that.

Response: Thanks for the kind suggestions. There are a few reports about trifluoroethanol and difluoroethanol radicals, but the synthetic application of these radicals is scarce, especially for the carbon radical derived from difluoroethanol. We added relative papers as refs 30-32 in the revised manuscript. We agree with Reviewer 3 that these radicals look like nucleophilic radicals. Although the synthetic application of these radicals are limited, Ref 30 disclosed that trifluoroethanol radical could add to highly electrophilic hexafluoro-2-butyne. Moreover, the substrates in our manuscript are electron-deficient olefins, and our results in Figure 5 shows that more electro-deficient substrate afforded higher yield (for example, 9ac, 43% yield; 9ae, 70% yield). So we think that calculation is not mandatory to support these radicals are nucleophilic.

- Why in some cases Mn(II)/TBHP conditions are used and in other cases Mn(III)/TBPB conditions, for instance Fig. 6 vs Fig. 7? or also in Fig. 8?

Response: Thanks for the comments. Both Mn(II)/TBPB and Mn(III)/TBPB are effective conditions. In order to test the generality of both conditions, we tested some substrates with Mn(II)/TBPB and others with Mn(III)/TBPB in the allylation reactions (Figure 5). For the

alkylation reaction, we tested Mn(II)/TBPB conditions (Figure 6). For alkenylation reactions, we tested Mn(III)/TBPB conditions. For the reactions to transfer difluoroethanol unit, we used similar conditions as that in the reactions to transfer trifluoroethanol unit. We believe that people can use both conditions to get synthetically useful yield for the reactions shown in this manuscript. We stated this information in page 3 of the manuscript (highlighted in yellow).

- On page 7, in the conclusion, the sentence "it is worthy to note that the reaction pathways of radical ..." should be changed. As rationalized in the SI, the three pathways have different mechanistic manifolds.

Response: Thanks for the suggestion. The sentence "It is worthy to note that the reaction pathways of radical III in the allylation, alkylation and alkenylation reactions are probably different..." was changed to "It is worthy to note that the reaction pathways in the allylation, alkylation and alkenylation reactions are probably different ..."

REVIEWERS' COMMENTS:

Reviewer #3 (Remarks to the Author):

The points I raised have been addressed on a satisfactory basis. This revised version is now OK to me for publication.